# Ammonium Glycyrrhizinate Prevents Apoptosis and Mitochondrial Dysfunction Induced by High Glucose in SH-SY5Y Cell Line and Counteracts Neuropathic Pain in Streptozotocin-Induced Diabetic Mice

**DOI:** 10.3390/biomedicines9060608

**Published:** 2021-05-26

**Authors:** Laura Ciarlo, Francesca Marzoli, Paola Minosi, Paola Matarrese, Stefano Pieretti

**Affiliations:** 1National Center for Drug Research and Evaluation, Istituto Superiore di Sanità, Viale Regina Elena 299, 00161 Rome, Italy; laura.ciarlo@iss.it (L.C.); francesca.marzoli@iss.it (F.M.); paola.minosi@iss.it (P.M.); 2National Center for Gender-Specific Medicine, Istituto Superiore di Sanità, Viale Regina Elena 299, 00161 Rome, Italy

**Keywords:** ammonium glycyrrhizinate, diabetic peripheral neuropathy, mitochondria, streptozotocin-induced diabetic mice, high glucose

## Abstract

Glycyrrhiza glabra, commonly known as liquorice, contains several bioactive compounds such as flavonoids, sterols, triterpene, and saponins; among which, glycyrrhizic acid, an oleanane-type saponin, is the most abundant component in liquorice root. Diabetic peripheral neuropathy is one of the major complications of diabetes mellitus, leading to painful condition as neuropathic pain. The pathogenetic mechanism of diabetic peripheral neuropathy is very complex, and its understanding could lead to a more suitable therapeutic strategy. In this work, we analyzed the effects of ammonium glycyrrhizinate, a derivate salt of glycyrrhizic acid, on an in vitro system, neuroblastoma cells line SH-SY5Y, and we observed that ammonium glycyrrhizinate was able to prevent cytotoxic effect and mitochondrial fragmentation after high-glucose administration. In an in vivo experiment, we found that a short-repeated treatment with ammonium glycyrrhizinate was able to attenuate neuropathic hyperalgesia in streptozotocin-induced diabetic mice. In conclusion, our results showed that ammonium glycyrrhizinate could ameliorate diabetic peripheral neuropathy, counteracting both in vitro and in vivo effects induced by high glucose, and might represent a complementary medicine for the clinical management of diabetic peripheral neuropathy.

## 1. Introduction

Glycyrrhiza glabra is a perennial plant that has been used in traditional Chinese medicine for thousand years and recently also in Europe [1]. Glycyrrhiza glabra was considered in the world as “the grandfather of plants”. Many countries used this plant in medicine for its pharmacological properties. Traditionally, this plant was used for treating asthma, hoarseness of voice, cough, and lung diseases. It was also used as a remedy for diseases of liver, heart palpitation, and angina. Its use has also been suggested in the treatment of bladder and kidney pain, kidney stones, fever, neuralgia, skin, and eye diseases [2].

Glycyrrhizic acid or glycyrrhizin is the main active component extracted from the glycyrrhiza root, this compound consisting of a triterpenoid pentacyclic glucoside [3]. Glycyrrhizic acid is an amphiphilic molecule: the hydrophilic part is represented by the glucuronic acid residues, and the hydrophobic part is the glycyrrhetic acid residue [3]. Glycyrrhizin has long been known as a compound with many biological effects: anti-inflammatory, antiulcer, antianaphylaxis, antioxidant, immunoregulatory, membrane stabilization, antiviral, and anticancer activities [4]. In particular, the anti-inflammatory activity of glycyrrhizic acid has been well studied; several in vitro studies have shown that glycyrrhizic acid can inhibit the production of the most proinflammatory cytokine such as TNF-α, interleukins IL-1β, and IL-6 [5]. Sun et al. showed that glycyrrhizic acid inhibits the synthesis of nitric oxide and some inflammatory cytokines in microglial cells treated with LPS [6]. The anti-inflammatory activity lies in its particular conformation, which consists of a pentacyclic triterpenic structure very similar to that of the well-known glucocorticoids. 

Two types of drugs can be used to treat pain in the chronic inflammatory diseases: nonsteroidal anti-inflammatory drugs (NSAIDs) and opioids. The treatment with NSAIDs for a long time often leads to several side effects, such as gastrointestinal lesions and nephrotoxicity [7]. In the same vein, the use of opioid induces respiratory depression, tolerance, and physical dependence [8]. 

Diabetes mellitus is a chronic metabolic disease characterized by a persistent excess of glucose in the blood, known as hyperglycemia. This condition is due to insufficient insulin production that may progress over time, leading to an insulin resistance condition. The major complications due to the onset of chronic diabetes are neuropathy, retinopathy, and nephropathy [9]. Half of the patients with diabetes suffer from peripheral neuropathy (DPN), leading to painful conditions as neuropathic pain negatively affects the quality of life. The pathogenetic mechanism of DPN is very complex, and its understanding could allow the implementation of a more suitable therapeutic strategy [10].

Chronic hyperglycemia appears to be one of the events responsible for the development of DPN. In fact, hyperglycemia induces an increase in the production of mitochondrial and cytoplasmic reactive oxygen species (ROS) while also inhibiting antioxidant defenses. This generates a vicious cycle in which mitochondrial dysfunction induces more ROS production leading a progressive decline of sensory neuron function and the loss of peripheral innervation that characterizes DPN [11,12]. Neuronal mitochondrial dysfunction and oxidative stress therefore represent key determinants in DPN [13]. 

The main parameter that influences the ROS generation in mitochondria is represented by the mitochondrial membrane potential (MMP) [14,15], and a significant production of ROS has been observed in mitochondria with more than >140 mV of MMP [16]. On the other hand, the increase in MMP could be the result of ATP synthase inhibition [17]. In fact, it has actually been demonstrated that mitochondria increase ROS production when they are not synthesizing ATP [18]. In the early phase of apoptosis, ROS generation is paralleled by a hyperpolarization of the mitochondrial membrane, followed by mitochondrial membrane depolarization in the execution phase [19]. Maintenance of shape and morphology of mitochondria, required for their normal functions, is regulated by a balance of constitutive fission and fusion dynamic processes. This remodeling has great relevance in both cell life and death, being the mitochondrial network involved in all activities, including proliferation, differentiation, and senescence, as well as cell death [20]. The oxidative stress plays an important role as inductor of mitochondrial fission [21], and several studies seem to indicate that hyperglycemia per se can also stimulate mitochondrial fragmentation [22].

Currently, there are no effective treatments for DPN apart from glycemic control. Saad et al. in their study showed that a new therapeutic strategy, which acts at the level of mitochondrial metabolism, could improve the management of DPN [23]. 

In this work, we analyzed the effects of ammonium glycyrrhizinate (AG) on a model used for studies on diabetic neuropathy, i.e., neuroblastoma cells line SH-SY5Y [24], and then, we validated in vivo its ability to counteract DPN in streptozotocin-induced diabetic mice.

## 2. Materials and Methods

### 2.1. In Vitro Experiments

#### 2.1.1. Cells and Treatments

The human neuroblastoma cell line SH-SY5Y was cultured in Dulbecco’s modified Eagle’s medium (DMEM) (Sigma-Aldrich, St. Louis, MO, USA) containing 4500 mg/L glucose, sodium pyruvate, and sodium bicarbonate; 10% fetal bovine serum, and penicillin and streptomycin at 100 U/mL penicillin and 100 mg/mL streptomycin (Sigma-Aldrich, St. Louis, MO, USA), and kept at 37 °C in a humidified 5% CO_2_ incubator. SH-SY5Y cells were obtained from ATCC, and all experiments were carried out up to 12 passages. In total, 80,000 cells/well were seeded on 12-well tissue culture plates. After 24 h, cells were treated with different concentrations of D-Glucose (GLU, Sigma-Aldrich, St. Louis, MO, USA) (75, 100, 150, 200, 250, and 300 mM) for 24, 48, and 72 h to find out the maximum concentration that induce cytotoxicity. At the same time, different concentrations of ammonium glycyrrhizate (AG, Sigma-Aldrich, St. Louis, MO, USA) (200 μg/mL) were used to select the concentration able to counteract the effects of high glucose. As control, we used cells treated with mannitol (MAN, Sigma-Aldrich, St. Louis, MO, USA), an osmotic sugar alcohol that is metabolically inert in humans, at the same concentration as GLU.

#### 2.1.2. Cell Death Assays

Quantitative evaluation of apoptosis was performed by a double-staining flow cytometry method by using fluorescein isothiocyanate (FITC)-conjugated Annexin V/PI apoptosis detection kit according to the manufacturer’s protocol (BioVision, Inc. Milpitas, CA, USA). Alternatively, cell death was analyzed after cell staining with Calcein-AM (Molecular Probes, Eugene, OR, USA). All samples were acquired and analyzed with a FACScalibur cytometer (BD Biosciences, San Jose, CA, USA). At least 20,000 events were acquired. Data were recorded and statistically analyzed by a Macintosh computer using CellQuest software (BD Biosciences, San Jose, CA, USA).

#### 2.1.3. Mitochondrial Membrane Potential

The mitochondrial membrane potential of controls and treated cells were studied by using 5-5′,6-6′-tetrachloro-1,1′,3,3′-tetraethyl benzimidazole-carbocyanine iodide probe (JC-1; Molecular Probes, Eugene, OR, USA), as described [25]. In line with this method, living cells were stained with 10 μM of JC-1. Tetramethylrhodamine ester 1 μM (TMRM; Molecular Probes, Eugene, OR, USA) was also used to confirm data obtained by JC-1 (Appendix A).

#### 2.1.4. Western Blot Analysis

The samples were subjected to sodium-dodecyl sulphate polyacrilamide gel electrophoresis (SDS-PAGE) under standard protocols. Briefly, each sample was lysed in Ripa buffer (50 µm Tris-HCL pH 7.4; 1% NP40; 0.5% Na—Deoxycholate; 0.1% SDS; 150 mM NaCl; 2 mM EDTA; 50 mM NAF) with protease and phosphatase inhibitor mixture. Protein content was determined by Bradford assay, and 40 µg of proteins were loaded. The proteins were electrophoretically transferred to the nitrocellulose membranes (GVS Life Sciences, GVS North America, Sanford, ME, USA) and probed with the following antibodies: MAb anti-DRP1 (BD Biosciences, San Jose, CA, USA), anti-hFis1 polyclonal antibody MAb anti-OPA1 (BD Biosciences, San Jose, CA, USA), PAb anti-MFN2 (Cell Signaling Technology, Inc., Danvers, MA, USA), for mitochondrial investigation; MAb anti-HMGB1 (Sigma-Aldrich, St. Louis, MO, USA), MAb anti-NFκB p65 (Santa Cruz Biotechnology Inc., Dallas, TX, USA) for inflammation investigation. Bound antibodies were visualized with horseradish peroxidase (HRP)-conjugated antirabbit IgG or antimouse IgG (Jackson Immuno Research Laboratories, Baltimore Pike West Grove, PA, USA) and immunoreactivity assessed by chemiluminescence reaction, using the ECL Western detection system (Millipor, Darmstadt, German). Densitometric scanning analysis was performed by Chemidoc (Bio-Rad, Hercules, CA, USA). To standardize, MAb anti-Tubulin (TUB, Sigma-Aldrich, St. Louis, MO, USA), PAb anti-Actin (Sigma-Aldrich, St. Louis, MO, USA), or anti-GAPDH (Sigma-Aldrich, St. Louis, MO, USA) were used for observations.

#### 2.1.5. Immunofluorescence Analysis

Control and treated cells were fixed with 4% paraformaldehyde (Carlo Erba, Milan, Italy) and then permeabilized by 0.5% Triton X-100 (Sigma-Aldrich, St. Louis, MO, USA). After washings, cells were incubated with PAb to TOM20 (Santa Cruz Biotechnology Inc., Dallas, TX, USA). After washings, cells were incubated with antirabbit AlexaFluor 594-conjugated (Termo Scientific, Rockford, IL, USA) for additional 1 h at 37 °C. All samples were counterstained with Hoechst 33342 (Termo Scientific, Rockford, IL, USA) and mounted with glycerol-PBS (2:1). The images were acquired by intensified video microscopy (IVM) with an Olympus fluorescence microscope (Olympus Corporation of the Americas, Center Valley, PA, USA), equipped with a Zeiss charge-coupled device (CCD) camera (Carl Zeiss, Oberkochen, Germany).

#### 2.1.6. Morphometric Analysis

Quantitative evaluations of mitochondrial fragmentation were carried out by evaluating at least 50 cells at the same magnification (×1300). Morphometric analysis was performed by using the ImageJ to measure the average mitochondrial area. To this purpose, red–green–blue (RGB) images were processed using a custom-written ImageJ macro containing plugins that calculate the average area of the mitochondrial particles throughout the cell cytoplasm using the outlines algorithm of the “analyze particles” function. The macroassisted algorithm was set to measure all particle sizes larger than the background pixelation but smaller than the average nuclear size. The average mitochondrial area is expressed as pixel^2^.

### 2.2. In Vivo Experiments

#### 2.2.1. Animals

All experiments were performed according to Legislative Decree 26/14, which implements the European Directive 2010/63/UE on laboratory animal protection in Italy, and were approved by the Service for Biotechnology and Animal Welfare of the Istituto Superiore di Sanità and authorized by the Italian Ministry of Health (approval code 198/2013-B). CD-1 mice (Charles River, Sant’Angelo Lodigiano LO, Italy) were housed in colony cages (seven mice per cage) under standard conditions of light, temperature, and relative humidity for at least 1 week before the start of experimental sessions. Food and water were available ad libitum.

#### 2.2.2. Streptozotocin-Induced Diabetes

It was reported that a high dose of streptozotocin (STZ, Sigma Aldrich, St. Louis, MO, USA) is directly toxic to pancreatic β-cells, rapidly causing diabetes, with blood glucose levels of >500 mg/dl within 48 h in mice. Thus, STZ dissolved in saline was administered at 200 mg/kg (1.0 mL/100 g) by a single intraperitoneal (i.p.) injection to induce diabetes [26]. Control mice were injected with vehicle alone. Measures of blood glucose were performed using a One Touch Basic blood glucose monitoring system (LifeScan Italy S.R.L, Milan, Italy) to ensure hyperglycemia. Body weight was also monitored. Only mice with blood glucose concentration exceeding 500 mg/dL were considered diabetic and used for the study. Notably, 15, 17, and 19 days after the STZ administration, diabetic mice received i.p. injection of saline (10 mL/kg, control mice) or AG in saline (50 mg/kg, 10 mL/kg).

Paw thermal withdrawal latency (PWL) was used to measure thermal hyperalgesia and performed by using an infrared generator (code 7360, Ugo Basile, Gemonio, VA, Italy). Mice were gently restrained using a glove, and after placing the mouse footpad in contact with the radiant heat source paw, withdrawal latency was measured. A timer initiated automatically when the heat source was activated, and a photocell stopped the timer when the mouse withdrew its hind paw. An intensity of 30 and a cut-off time of 15 s were used of the heat source on the plantar apparatus to avoid tissue damage. The PWL, in terms of seconds, of each animal in response to the plantar test was determined [27]. Baseline paw thermal withdrawal latencies were determined before saline or AG administration.

### 2.3. Data Analysis and Statistics

#### 2.3.1. In Vitro Data

Collected data analysis was carried out by using 2-way ANOVA test for repeated samples corrected for multiple comparisons by the Bonferroni procedure. All data reported in this paper were verified in at least 3 different experiments performed in triplicate and reported as mean ± standard deviation (SD). Flow cytometry analyses were performed by using a FACSCalibur flow cytometer (BD Biosciences, San Jose, CA, USA) equipped with a 488 argon laser and with a 635 red diode laser. At least 20,000 events/sample were acquired and analyzed using the Cell Quest Pro software (BD Biosciences, San Jose, CA, USA).

#### 2.3.2. In Vivo Data

Statistically significant differences between groups were measured with an analysis of variance (2-way ANOVA) followed by Sidak’s or Dunnett’s post hoc comparisons test, when the comparison was restricted to two groups. GraphPad Prism 6.0 software (San Diego, CA, USA) was used to analyze the data.

## 3. Results

### 3.1. AG Counteracted High Glucose-Induced Cell Death

One of the causes of DPN is death of Schwann cells due to prolonged exposure to high glucose and consequent oxidative stress. On this basis, we investigated the effect of GLU by using SH-SY5Y neuroblastoma as a model cell line. First of all, to determine the effect of high glucose on viability of SH-SY5Y cells, we treated cells with increasing concentrations of GLU or MAN (0–300 mM) for different time points (24, 48 and 72 h). We observed that, starting from 48 h, GLU was able to induce an appreciable cytotoxic effect. The results obtained at this timepoint using Calcein-AM showed that the percentage of dead cells increased along with increasing concentrations of GLU, being about 30% at a concentration of 300 mM (Figure 1A). Importantly, percentage of cell death induced by MAN was significantly lower than that induced by the same concentration of GLU and not significantly different from control untreated cells (Figure 1A). The antiapoptotic and anti-inflammatory effect of AG has been reported in various contexts [28]. Thus, we tested two different concentrations (500 μg/mL and 1000 μg/mL) of AG to verify its ability to inhibit the cytotoxic effects induced by 300 mM GLU (Figure 1B). We observed a protective effect exerted by AG already at 500 μg/mL, which became significantly more marked by increasing the concentration of AG to 1000 μg/mL. On the basis of these preliminary experiments, we selected 300 mM as the concentration of GLU inducing an appreciable cytotoxic effect (also indicated as high glucose through the text, HG) and 1000 μg/mL as the concentration of AG capable of significantly inhibiting it (Figure 1B). Note that both fall within the range of concentrations used, as reported in the literature [29,30].

### 3.2. AG Counteracted HG-Induced Apoptosis and Mitochondrial Alterations

Through flow cytometry analysis after double cell staining with Annexin V (AV)/propidium iodide (PI), we analyzed apoptosis in SH-SY5Y neuroblastoma cells growing in HG conditions for 48 h. We found that HG induced cell death by apoptosis in about 30% of SH-SY5Y cells and that 1000 μg/mL AG was able to almost completely inhibit HG-induced apoptosis (Figure 2A). Neither MAN (considered as an internal control) nor AG-treated cells showed any significant difference when compared to untreated control cells. It is known that hyperglycemia induces mitochondrial dysfunction [31]. We therefore analyzed the MMP by flow cytometry, after cell staining with JC-1 probe. HG induced a significant increase in the percentage of cells with high MMP, i.e., hyperpolarized mitochondria (boxed area, in Figure 2B). According to apoptosis data, AG was able to significantly (*p* < 0.01) reduce the percentage of cells with high MMP induced by HG, and MAN did not induce any alteration of MMP (boxed areas in Figure 2B). Overlapping results were obtained using TMRM as a probe for the study of the MMP (Appendix A).

The accumulation of evidence indicates that the mitochondrial fragmentation and fission represent important contributing factors to alterations of mitochondrial membrane and ATP production [32,33]. Thus, we also evaluated the mitochondrial network organization by immunofluorescence analysis after cell staining with antimitochondrial import receptor subunit TOM20 (red) and counterstaining with Hoechst (blue) in cells grown under HG conditions. We found that GLU induced mitochondrial fragmentation (Figure 3A), which is normally associated with dysfunctional mitochondria [34]. The administration of AG to cells growing under HG restored normal mitochondrial morphology, as also revealed by morphometric analysis performed by using ImageJ to measure average mitochondrial area in cells stained with an anti-TOM20 antibody (Figure 3B).

Mitochondria form a highly interconnected and dynamic network and provide necessary ATP for the smooth functioning of cells [35]. Mitochondrial fragmentation due to fission process is promoted by the activation of dynamin-related protein 1 (DRP1) [36], which, under stress conditions, is recruited in mitochondria [37]. In mammals, at least two proteins, the dynamin-related protein DRP1 and the mitochondrial outer membrane protein hFIS1, participate in mitochondrial fission. The opposite process, mitochondrial fusion, is regulated by various proteins, among which are mitofusin 2 (MFN2) which mediates fusion of outer membrane and the optic atrophy protein 1 (OPA1) which carries out fusion of inner membrane [38]. Since mitochondria fusion and fission are strictly related, the activation of one of these processes leads to the inhibition of the other [39]. Thus, we evaluated DRP1, hFIS1, MFN2, and OPA1 by Western blot analysis. HG induced an increase in the expression levels of both DRP1 and hFIS1 proteins, while in cells subjected to the simultaneous administration of GLU and AG, we did not observe any significant difference from control untreated cells (Figure 4A,B). As far as proteins involved in the fusion process were concerned, HG seemed not to influence significantly the expression of MFN2 but dramatically reduced the expression of OPA1. This reduction was effectively countered by treatment with AG (Figure 4C).

### 3.3. AG Counteracted Inflammation Induced by HG

Diabetic patients have been found to increased HMGB1 and RAGE levels [40,41]. HMGB1 is normally expressed in the nucleus. However, following signals of stress, injury, or tissue damage, the protein was released into the extracellular space. HMGB1 can bind the Toll-like receptor 4 (TLR4) and the receptor for advanced glycation end-products (RAGE), leading to increased inflammation commonly through nuclear factor kappa beta (NFκB) [42]. On these bases, we investigated the possible anti-inflammatory activity of AG in our cell model analyzing by Western blot HMGB1, which is a ubiquitous nuclear protein that promotes inflammation when extruded from the cell after stress, damage, or death, and p65-nuclear factor kappa-light-chain-enhancer of activated B cells (NFκB), which is a critical regulator of immune and inflammatory responses. According to the literature’s data, HG condition induced a significant increase in the expression levels of both HMGB1 and NFκB. Administration of AG in cells subjected to HG effectively decreased the level of both proinflammatory proteins (Figure 5).

### 3.4. AG Induced Anti-Hyperalgesic Effect in Diabetic Mice

To test the efficacy of AG in preventing or mitigating diabetic neuropathy induced by hyperglycemia in vivo, we used STZ as inducers of diabetes in mice. The results of experiments performed in diabetic mice are reported in Figure 6. STZ resulted in increased thermal hyperalgesia—a reduction in the withdrawal threshold to a noxious stimulus—when compared to baseline recorded before diabetes as observed 13 days after STZ treatment. When AG was first administered 15 days after STZ, a nonsignificant increase in paw withdrawal latencies was observed (Figure 6). AG was administered again at days 17 and 19 after STZ injection, and at this time, a strong increase in paw latency was recorded. Thus, our data demonstrated that a short-repeated treatment with AG is able to induce an anti-hyperalgesic effect in diabetic mice.

## 4. Discussion

According to the World Health Organization, diabetes is defined as a chronic metabolic disease characterized by an incorrect glucose metabolism resulting from defects in insulin secretion or insulin action. Diabetic peripheral neuropathy (DPN) is one of the most common complications of diabetes, affecting at least 50% of patients with diabetes during their lifetime [43]. For the treatment of DPN, clinical guidelines recommend symptomatic treatments with different classes of drugs, which include opioids, the γ-aminobutyric acid analogues, and antidepressants [23]. Unfortunately, these drugs induce many adverse effects on the level of several organs; hence, it is necessary to identify new strategies for the management of DPN and DPN-associated pain, both in terms of prevention and treatment. Many clinical studies have been carried out with unsuccessful results, and consequently, the therapeutic advances have not been noteworthy. In fact, as of now, the only recommendation for preventing or slowing the progression of peripheral neuropathy is to maintain close glycemic control. It has also been suggested that pharmacological activation of protective neuronal circuits may be useful for increasing tolerance to the hyperglycemic condition and for promoting neuronal recovery [44].

Although the pathogenic mechanisms of DPN are still not fully understood, the multifactorial nature of the pathological changes induced by the lack of glycemic control is now clear [45]. One of the mechanisms that lead to the damage of the nervous tissue in DPN is activated in the state of hyperglycemia that triggers the increase in mitochondrial and cytoplasmic ROS/NS production. Mitochondria are particularly abundant at neuronal synapse levels where they play an important role in maintaining calcium homeostasis, and a strong link between neuronal dysfunction and mitochondrial dysfunction has also been observed [46]. The mitochondria constitute a dynamic network that, through fusion and fission processes, maintain their efficiency and distribution within the cell on the basis of energy needs. Furthermore, the mitochondrial transport chain has been shown to contribute significantly to the neuropathic pain, whose pathogenesis would therefore be strongly ATP dependent [47]. The centrality of mitochondrial damage in the etiopathogenesis of DPN was also suggested by the positive correlation between the stage of the disease and the relationship between whole blood and plasma mtDNA levels in patients with diabetes [48]. In mammals, mitochondrial fragmentation due to fission process is promoted by the activation of DRP1 protein [36], which, under stress conditions, is recruited in mitochondria [37] and by the mitochondrial outer membrane protein hFIS1. Various proteins, among which MFN2 and OPA1, regulate mitochondrial fusion [38]. The balance between fission and fusion is crucial for cell fate, and despite the complex etiology of neuropathic pain, a fundamental role has recently been recognized to the excess of mitochondrial fission as a possible early cause of neuronal loss due to apoptotic death [12,49]. In accordance with the central role played by the mitochondrion in the pathogenesis of DPN, we observed in SH-SY5Y neuroblastoma cells subjected to hyperglycemic stress an alteration of the MMP (i.e., hyperpolarization), a mitochondrial fragmentation accompanied by the increase in the expression of DRP1 and hFIS and the decrease in MFN2, and, finally, a significant increase in apoptosis. Moreover, as also reported by numerous recent studies carried out both in vitro and in vivo [50,51,52,53], we found that SH-SY5Y cells subjected to HG conditions increased the expression of HMGB1 and NFκB. HMGB1, when released by the cell actively or passively due to its death, can interact with the receptor for advanced glycation end products (RAGE) or with Toll-like receptor 4 (TLR4) triggering, by the activation of NFκB, an inflammatory response. Although a strong activation of the inflammatory pathways contributes to diabetic neuropathy, and in the development and maintenance of neuropathic pain, it has nevertheless been shown that normalization of the inflammatory state is per se not sufficient to revert the neuropathic state. In fact, only the concomitant re-establishment of mitochondrial bioenergetic conditions would seem able to attenuate DPN [54].

Herbs have been used throughout the world in medical practice, for example in traditional Chinese medicine [55] and Ayurvetica [56]. However, due to their complexity, it is very difficult to standardize the composition and production of herbal medicines. On the other hand, the bioactive molecules contained in plants can be isolated and studied by the same methods used for drugs in allopathic medicine. For example, Arauna and coworkers reported that many natural bioactive compounds have the ability to regulate oxidative phosphorylation, production of reactive oxygen species, and mitochondrial dysfunction [57]. Here, we analyzed the effects of AG, a salt derived from the natural triterpene glycoconjugate of the root of licorice, which is the main active compound known to exhibit pharmacological activities—including anti-inflammatory and antioxidative ones [58,59]—in preventing the damage caused by HG in the SH-SY5Y cell line. This cell line is commonly used as an in vitro model for the study of diabetic neuropathy, because, when treated with HG, it shows similar alterations to those observed in dorsal root ganglion (DRG) neurons and Schwann cells [24,60]. First of all, we observed that AG was able to attenuate the cytotoxic effect of high glucose (300 mM, 48 h) by inhibiting cell death. Going deeper into the mechanisms underlying this protective effect, we found that cell treatment with AG completely prevented the alterations of the mitochondrial membrane potential and the mitochondrial fragmentation induced by high concentrations of glucose. Although the anti-inflammatory activity of AG has already been documented in experimental models of diabetes [61,62,63], to the best of our knowledge, the effects of this natural compound at the mitochondrial level are reported for the first time in this work.

Too often the encouraging results obtained on simplified in vitro models are not matched by more complex in vivo models. Although no mouse model exactly reproduces DPN observed in humans, streptozotocin-induced diabetic mice are currently considered an excellent model for preclinical studies [64], as they develop primarily distal axon loss, systemic injury of the peripheral nervous system, and altered interactions with Schwann cells, which are recognized as typical features of DPN [65]. Importantly, in this animal model, we demonstrated that a short-repeated treatment with AG was able to contrast significantly thermal hyperalgesia, known to be associated with increased HMGB1 expression in DRG, as recently reported in Zucker diabetic rat and ob/ob mice [61]. On the other hand, AG is also able to reduce or suppress nociception in other pain models [66] and the persistent endogenous release of HMGB1 by sensory neurons may be a potent, physiologically relevant modulator of neuronal excitability as observed in tibial nerve injured neuropathic rat [11]. On the other hand, glycyrrhizin bound directly to both HMG boxes of HMGB1 and inhibited its chemoattractant and mitogenic activities [67]. Recently, many studies have investigated the mechanisms of HMGB1 inhibition by AG in a wide number of HMGB1-involved diseases, and it has been demonstrated that AG inhibited extracellular HMGB1 cytokine activity, and protected spinal cord, liver, brain, and myocardium against ischemia–reperfusion (I/R)-induced injury in experimental animals [68].

Considering the absence of cytotoxicity, even at high concentrations, and the good pharmacological tolerability in rodents and humans after acute or subchronic treatment [69], AG may represent a complementary medicine in the clinical management of DPN with the added advantage of providing a multitarget effect on the various etiological factors underlying the pathophysiology of DPN, such as inflammation and mitochondrial damage.

## Figures and Tables

**Figure 1 biomedicines-09-00608-f001:**
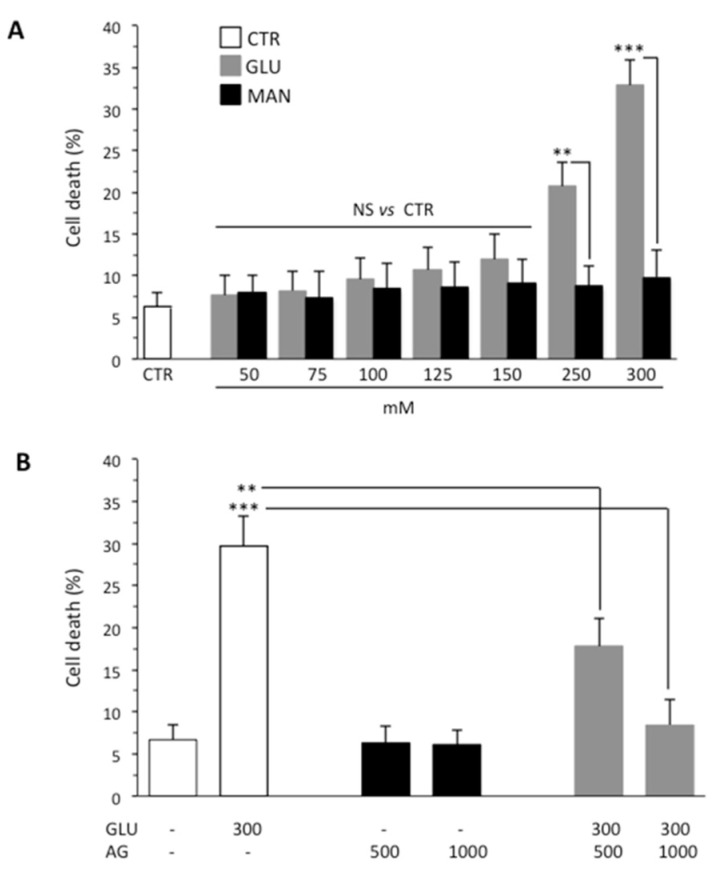
Flow cytometry analysis after staining with Calcein-AM (which is retained in the cytoplasm of live cells). (**A**) SH-SY5Y neuroblastoma cells were analyzed after treatment with increasing concentrations (0–300 mM) of GLU or MAN (used as an osmolarity control) for 48 h. Numbers represent the percentage of Calcein-negative cells (dead cells). Results obtained from three independent experiments are reported as means ± SD. (**B**) SH-SY5Y cells tested after treatment with 300 mM GLU or MAN alone or in combination with two different concentrations (500 and 1000 μg/mL) of AG. Results obtained from four independent experiments are reported as means ± SD. ** is for *p* < 0.01, and *** is for *p* < 0.001.

**Figure 2 biomedicines-09-00608-f002:**
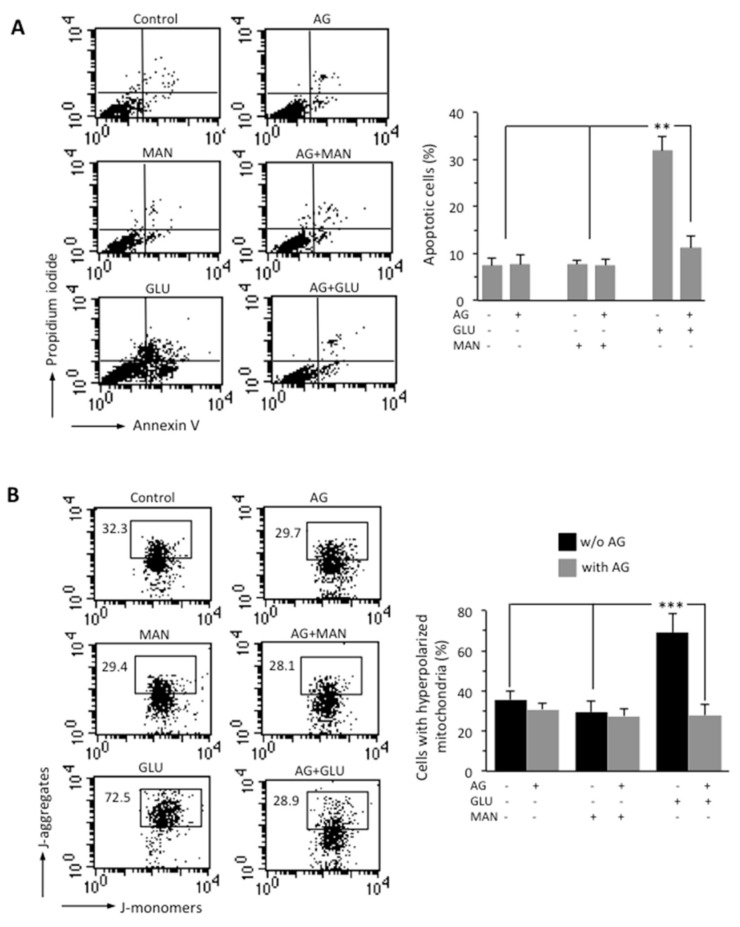
SH-SY5Y neuroblastoma cells were treated with 300 mM GLU or MAN alone or in combination with 1000 μg/mL AG for 48 h. (**A**) FACS analysis after double cell staining with Annexin V/PI. Left panels. Dot plots from a representative experiment are shown. Numbers represent the percentages of Annexin V-positive cells (bottom-right quadrant), Annexin V/PI double-positive cells (upper right quadrant), or PI-positive cells (upper left quadrant). Bar graph shows results obtained from four independent experiments, reported as means ± SD. (**B**) Left panels. Representative dot plots of the FACS analysis of MMP, performed by using JC-1. Numbers in the boxed areas indicate the percentage of cells with high MMP (hyperpolarized mitochondria). Bar graph shows the percentage of cells with hyperpolarized mitochondria obtained from three independent experiments and reported as percentage  ±  SD. ** is for *p* < 0.01, and *** is for *p* < 0.001.

**Figure 3 biomedicines-09-00608-f003:**
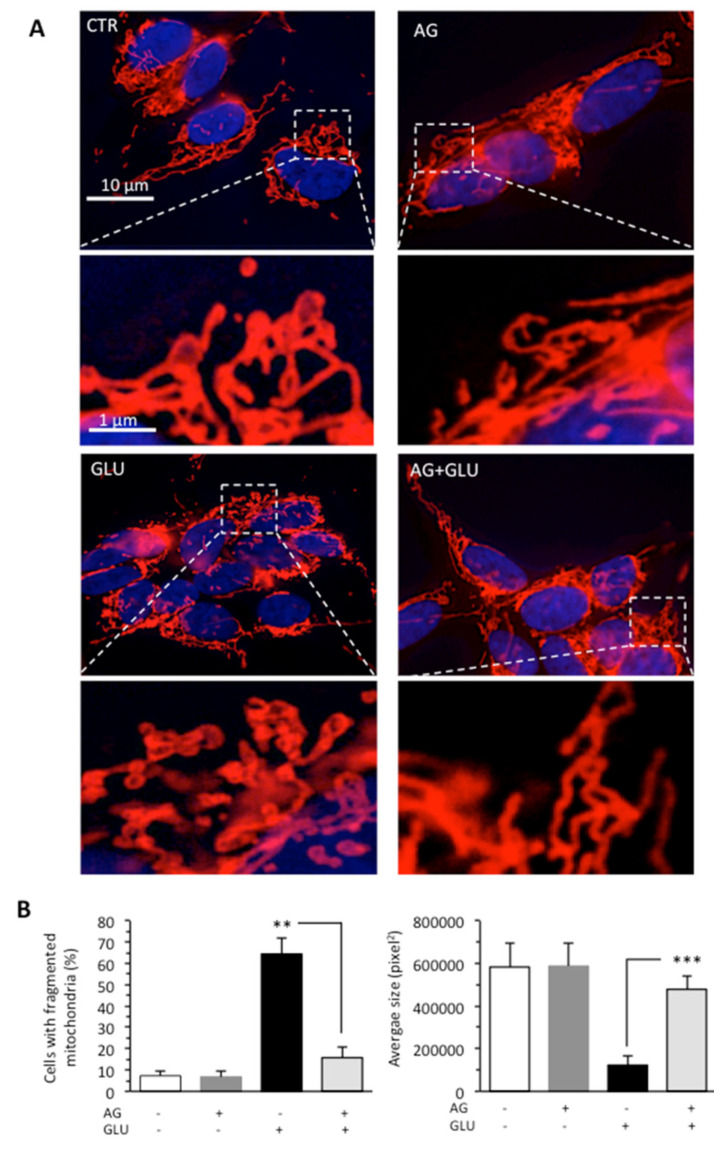
SH-SY5Y neuroblastoma cells were treated with 300 mM GLU or MAN alone or in combination with 1000 μg/mL AG for 48 h. (**A**) Representative micrographs obtained by IVM showing the mitochondrial network of SH-SY5Y neuroblastoma cells, untreated (CTR) or treated for 48 h with 100 mM GLU with or without AG, after staining with anti-TOM20 (red) and counterstaining with Hoechst (blue). In the bottom pictures, magnification of the boxed areas. (**B**) Left panel. Bar graph showing the percentage of cells in which fragmented mitochondria were observed. Right panel. Morphometric analysis performed by using the ImageJ software. In ordinate average mitochondrial area is expressed as pixel^2^. Data are reported as mean value  ±  SD of the results obtained by analyzing at least 50 cells for each sample. ** is for *p* < 0.01, and *** is for *p* < 0.001.

**Figure 4 biomedicines-09-00608-f004:**
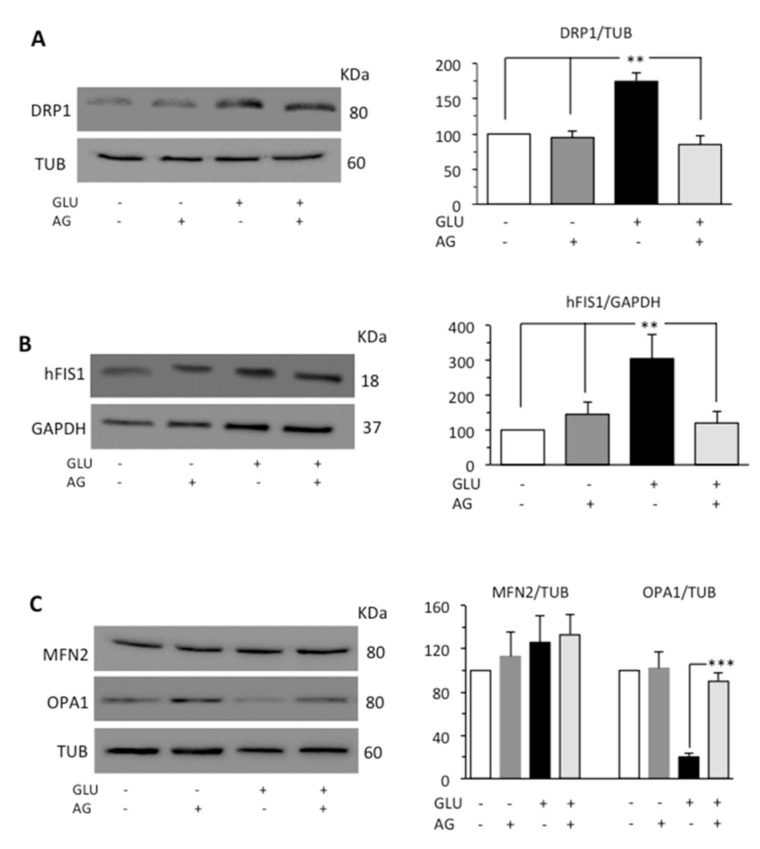
Western blot analysis of (**A**) DRP1 and (**B**) hFIS1, and (**C**) MFN2 and OPA1, involved in fission and fusion processes, respectively. Loading control was evaluated using anti-TUB or anti-GAPDH MAb. A representative experiment among the three is shown. Bar graphs on the right panel show densitometric analysis of the band density ratio of each protein relative to loading control. Mean ± SD from three independent experiments is shown. ** *p* ≤ 0.01 and *** *p* ≤ 0.001.

**Figure 5 biomedicines-09-00608-f005:**
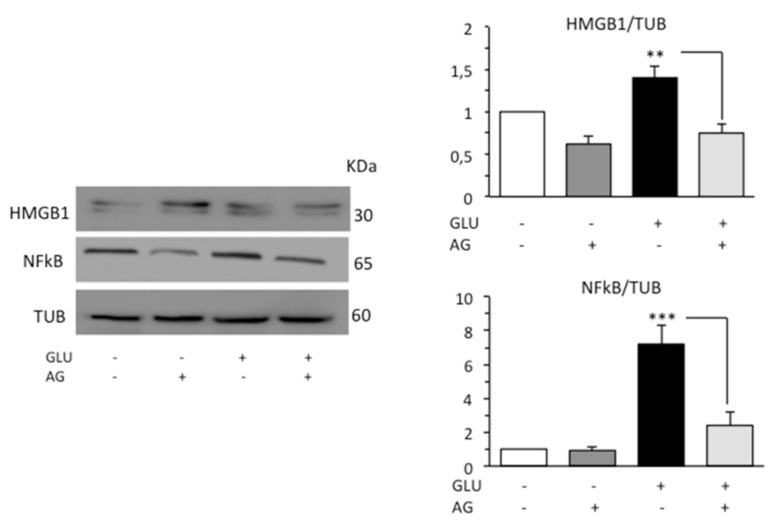
Western blot analysis of the inflammatory proteins HMGB1 and NFκB. Loading control was evaluated using anti-TUB. A representative experiment among the three is shown. Bar graphs on the right panels show densitometric analysis of the band density ratio of each protein relative to loading control. Mean ± SD from three independent experiments is shown. ** *p* ≤ 0.01 and *** *p* ≤ 0.001.

**Figure 6 biomedicines-09-00608-f006:**
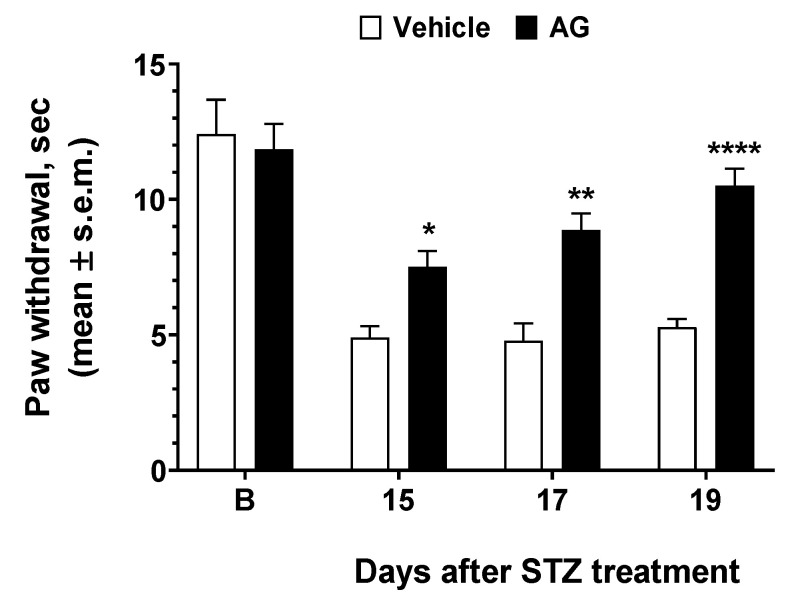
Effects induced by AG in Streptozotocin-induced diabetic mice. Paw withdrawal latency was recorded before STZ treatment (baseline latency, B) and three more times at 15, 17, and 19 thereafter. Vehicle (V, saline 10 mL/kg) and AG (50 mg/kg, 10 mL/kg) were administered three times at day 15, 17, and 19 after STZ injection. * is for *p* < 0.05, ** is for *p* < 0.01, and **** is for *p* < 0.001 vs V. *n* = 9.

## Data Availability

Data can be available upon request.

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
