# Peer review of "Ammonium Glycyrrhizinate Prevents Apoptosis and Mitochondrial Dysfunction Induced by High Glucose in SH-SY5Y Cell Line and Counteracts Neuropathic Pain in Streptozotocin-Induced Diabetic Mice"

_biomedicines, 2021, doi:10.3390/biomedicines9060608_

Round 1

Reviewer 1 Report

In this article, Laura Ciarlo et al. investigated the effects of ammonium glycyrrhizinate (AG) on diabetic peripheral neuropathy by in in vitro and in vivo approach and found that AG inhibited the high glucose-induced apoptosis of SH-SY5Y cells by preventing mitochondrial alterations. Moreover, AG ameliorated the hyperalgesia in streptozotocin-induced diabetic mice. The study is well designed and the founds includes important information for the development of anti-diabetic peripheral neuropathy drugs. I recommend this article be published. I have some comments and hope will help you to improve your manuscript.

1. In in vitro experiments, AG suppressed the high glucose-induced upregulation of DRP1 and hFIS1, which resulted in the prevention of mitochondrial disfunction. My question is, what is the mechanism of this effect? Do you have any insights into the receptors for AG?

2. AG suppressed streptozotocin-induced hyperalgesia. Do you think this is also due to anti-apoptosis effect, as in in vitro experiment? My interest is that the repeated administration of AG accelerated the recovery of hyperalgesia. This result may suggest the regeneration of neurons. I think that pathological analysis is important for further understanding. Did you perform it?

3. (Figure 2) In general, annexin+PI- and annexin+PI+ cells are counted for cell death, so I think annexin-PI+ cells are not necessary.

Author Response

We thank the reviewer 1 for comments and helpful suggestions. The answer to Reviewer 1 Report are following reported.

  1. In in vitro experiments, AG suppressed the high glucose-induced upregulation of DRP1 and hFIS1, which resulted in the prevention of mitochondrial disfunction. My question is, what is the mechanism of this effect? Do you have any insights into the receptors for AG?

It is well known that mitochondrial fragmentation due to fission process is promoted by the activation of dynamin-related protein 1 DRP1 and the mitochondrial outer membrane protein hFIS1. In our experiments, high glucose (HG) induced an increase of the expression levels of both DRP1 and hFIS1 proteins, and ammonium glycyrrhizinate (AG) was able to counteract these effects. HG also induced a significant increase of the expression levels of both HMGB1 and NFkB and AG administration decreased the level of both pro-inflammatory protein. Furthermore, AG bound to both HMG boxes of HMGB1 and inhibited its activities HMGB1 (for a review, please see Musumeci et al., Pharmacol Ther. 2014 Mar;141(3):347-57. doi: 10.1016/j.pharmthera.2013.11.001). Recently, it was reported that pharmacological inhibition of HMGB1 by glycyrrhizin prevented Drp1-mediated mitochondrial fission in a model of pulmonary arterial hypertension (Feng et al., Cell Prolif  2021 May 4;e13048. doi: 10.1111/cpr.13048). No data are reported in literature about AG receptors, and we think that the decrease of HMGB1 level and its inhibition induced by AG might explain the effects observed in our experiments. Some data about AG inhibition of HMGB1 activities are reported in the revised version of the manuscript. Please see, page 13 lines 429-435 in the revised manuscript.

  1. AG suppressed streptozotocin-induced hyperalgesia. Do you think this is also due to anti-apoptosis effect, as in in vitro experiment? My interest is that the repeated administration of AG accelerated the recovery of hyperalgesia. This result may suggest the regeneration of neurons. I think that pathological analysis is important for further understanding. Did you perform it?

No data are reported in the literature to our knowledge on the possible effects of AG on neuronal regeneration and investigating this possibility could be the basis for future work. In any case, we believe that AG counteracts the detrimental effects induced by high glucose on nervous cells involved in nociception, as reported in the discussion section of the manuscript.  

  1. (Figure 2) In general, annexin+PI- and annexin+PI+ cells are counted for cell death, so I think annexin-PI+ cells are not necessary.

Following the reviewer’s suggestion, we added the percentages relative to the two cell populations (annexin+PI- and annexin+PI+) and modified the Figure accordingly. Please, see Figure 2 in the revised version of the manuscript.

Reviewer 2 Report

The authors describe that ammonium glycyrrhizinate prevents apoptosis and mitochondrial dysfunction in in vitro experiments and that counteracts neuropathic pain in diabetic mice.

As the authors write in the manuscript, the mechanisms of neuropathic pain are very complicated. Several methods are conducted in animal experiments because there are no perfect experiments to evaluate analgesic effects of the target substance on neuropathic pain. However, in this manuscript thermal hyperalgesia is only tried. At least, one more animal experiment should be added. I think von Frey test is indispensable for analgesic effect evaluation. Then, only one dose (50mg/kg) is used. Why did the authors decide the dose in animal experiment? As a pharmacological manuscript, at least two doses are necessary to clarify the effects.

Typographical errors:

P6, last sentence

“MMP induce by HG” to “MMP induced by HG”

P7, Figure 2

Black bar; with AG and Grey bar; w/o AG

Are they opposite?

Author Response

We thank the reviewer 2 for comments and helpful suggestions. The answer to Reviewer 2 Report are following reported.

As the authors write in the manuscript, the mechanisms of neuropathic pain are very complicated. Several methods are conducted in animal experiments because there are no perfect experiments to evaluate analgesic effects of the target substance on neuropathic pain. However, in this manuscript thermal hyperalgesia is only tried. At least, one more animal experiment should be added. I think von Frey test is indispensable for analgesic effect evaluation. Then, only one dose (50mg/kg) is used. Why did the authors decide the dose in animal experiment? As a pharmacological manuscript, at least two doses are necessary to clarify the effects.

The antinociceptive effects of AG are already reported in literature using several models of pain, and glycyrrhizin administration also alleviated CFA-evoked mechanical allodynia and thermal hyperalgesia in inflammatory pain model of mice (Sun et al., Exp. Cell. Res. 2018 369, 112-119). At our knowledge, no data are reported in the literature on the AG effects on streptozotocin-induced diabetic neuropathy, and for the first time we described these effects. On the other hand, Thakur et al. described the antinociceptive effects of AG in Zucker diabetic rat (Thakur et al., Int J Mol Sci. 2020 Jan 30;21(3):881. doi: 10.3390/ijms21030881) using glycyrrhizin for five days a week for four weeks at a dose of 50 mg/kg per day, measuring mechanical and thermal hyperalgesia. In our previous work, we used AG at 100 and 50 mg/kg, and AG at the dose of 50 mg/kg was able to reduce nociception in several pain models (Maione et al., Molecules. 2019 Jul 4;24(13):2453. doi: 10.3390/molecules24132453). For the above reasons and in order to save animals, we decided to use AG at a dose of 50 mg/kg in only one pain model.

Typographical errors:

P6, last sentence

“MMP induce by HG” to “MMP induced by HG”.

Modified as suggested. Please, see page 6, line 253 of the revised manuscript.

P7, Figure 2

Black bar; with AG and Grey bar; w/o AG

Are they opposite?

We thank the reviewer for the tip and  we apologize for the inaccuracy. In the new version of the figure 2 the legend has been corrected.

Reviewer 3 Report

I have some comments (please see the attachment for details).

Author Response

We thank the reviewer 3 for comments and helpful suggestions. The answer to Reviewer 2 Report are following reported.

  1. The manuscript is not in the appropriate format, it's missing line numbers. Therefore, it's very difficult to comment specifically.

We apologize for the mistake. In the revised version of the manuscript, we report the line numbers.

  1. Please delete ‘.’ From the title.

Modified as you suggested.

  1. Page 2, please correct the spelling of cytokine.

Modified as you suggested. Please, see pag. 2, line 47 in the revised version of the manuscript.

  1. Please make the text Italic whenever required. For example, on page 2, in vivo.

Modified as you suggested through the entire revised manuscript.

  1. Please write CO2 instead of CO2.

Modified as you suggested. Please see page 3, line 101 in the revised version of the manuscript.

  1. Data obtained by JC‐1 (not shown). Why? Please include the data if you mention it.

In the Materials and Methods section we have specified the following: “Tetramethylrhodamine ester 1 μM (TMRM; Molecular Probes) was also used to confirm data obtained by JC‐1 (not shown)” Figure 2B shows the results obtained using JC-1. As requested by the reviewer, in the new Supplementary figure S1 we now include the results obtained using TMRM.

  1. (50 μm Tris-HCL pH 7.4; 1% NP40; 0.5% Na – Deoxycholate; 0.1 % SDS; 150 m M NaCl ; 2 mM EDTA; 50 mM NAF). Why they are written differently? Use similar.

Modified as you suggested. Please, see page 3, line 129 in the revised version of the manuscript.

  1. 1.4. mithocondrial investigation should be mitochondrial investigation.

Modified as you suggested. Please, see page 3, line 136 in the revised version of the manuscript.

  1. 1.4. MAb anti-NFkB p65 should be MAb anti-NFkB p65. Correct everywhere.

Modified as you suggested through the entire revised manuscript.

  1. 1.5. with anti-rabbit AlexaFluor 594-conjugated (Termo Scientific Rockford, IL, USA) for additional 1h at 37° C. Please mention secondary antibody. Termo should be Thermo.

Modified as you suggested. Please, see page 4, line 149 in the revised version of the manuscript.

  1. Figure 3, there μM should be μm.

Modified as you suggested. Please, see Figure 3 in the revised version of the manuscript.

  1. Figure 4, 5, the western blot background is higher. Please reduce the background for better visualization.

Modified as you suggested. In the new version of the Figure 4 and 5 we reduced the western blot background as much as possible.

Round 2

Reviewer 2 Report

I do not have any more comments.